# FosL1 Is a Novel Target of Levetiracetam for Suppressing the Microglial Inflammatory Reaction

**DOI:** 10.3390/ijms222010962

**Published:** 2021-10-11

**Authors:** Kouji Niidome, Ruri Taniguchi, Takeshi Yamazaki, Mayumi Tsuji, Kouichi Itoh, Yasuhiro Ishihara

**Affiliations:** 1Program of Biomedical Science, Graduate School of Integrated Sciences for Life, Hiroshima University, Hiroshima 739-8521, Japan; babufu@icloud.com (K.N.); tngc.2019@gmail.com (R.T.); 2Program of Life and Environmental Sciences, Graduate School of Integrated Sciences for Life, Hiroshima University, Hiroshima 739-8521, Japan; takey@hiroshima-u.ac.jp; 3Department of Environmental Health, University of Occupational and Environmental Health, Fukuoka 807-8555, Japan; tsuji@med.uoeh-u.ac.jp; 4Laboratory for Pharmacotherapy and Experimental Neurology, Kagawa School of Pharmaceutical Sciences, Kagawa Bunri University, Sanuki 769-219, Japan; itoh@kph.bunri-u.ac.jp

**Keywords:** levetiracetam, FosL1, inflammation, microglia, post-brain-insult epilepsy

## Abstract

We previously showed that the antiepileptic drug levetiracetam (LEV) inhibits microglial activation, but the mechanism remains unclear. The purpose of this study was to identify the target of LEV in microglial activity suppression. The mouse microglial BV-2 cell line, cultured in a ramified form, was pretreated with LEV and then treated with lipopolysaccharide (LPS). A comprehensive analysis of LEV targets was performed by cap analysis gene expression sequencing using BV-2 cells, indicating the transcription factors BATF, Nrf-2, FosL1 (Fra1), MAFF, and Spic as candidates. LPS increased AP-1 and Spic transcriptional activity, and LEV only suppressed AP-1 activity. FosL1, MAFF, and Spic mRNA levels were increased by LPS, and LEV only attenuated FosL1 mRNA expression, suggesting FosL1 as an LEV target. FosL1 protein levels were increased by LPS treatment and decreased by LEV pretreatment, similar to FosL1 mRNA levels. The FosL1 siRNA clearly suppressed the expression of TNFα and IL-1β. Pilocarpine-induced status epilepticus increased hippocampus FosL1 expression, along with inflammation. LEV treatment significantly suppressed FosL1 expression. Together, LEV reduces FosL1 expression and AP-1 activity in activated microglia, thereby suppressing neuroinflammation. LEV might be a candidate for the treatment of several neurological diseases involving microglial activation.

## 1. Introduction

Microglia function in immune surveillance in the brain and are activated by inflammatory stimulation, which causes their morphology to change from ramified to amoeboid. Activated microglia produce humoral factors such as inflammatory cytokines and remove foreign substances in the brain to protect neurons. On the other hand, excessive microglial activation causes neuroinflammation, which can injure the normal brain. In fact, neuroinflammation is considered a pathological mechanism of several neurological diseases, such as Alzheimer′s disease, cerebral stroke, and epilepsy [1,2,3]. Therefore, it is important to suppress abnormal microglial activation for the treatment or prevention of a number of neurological diseases.

Post-brain-insult epilepsy, such as post-traumatic epilepsy, post-stroke epilepsy, or post-status epilepticus (SE) epilepsy, accounts for approximately 20% of symptomatic seizures and 5% of all epileptic seizures [4,5]. Microglial activation and neuroinflammation, which are observed during epileptogenesis after brain injury, can play an important role in the onset of seizures [6,7]. Neuroinflammation has the possibility of decreasing the seizure threshold, which shifts the neural excitation/inhibition balance toward excitation by modulating neural channel activity and/or neurotransmitter uptake or release [8,9]. Proinflammatory cytokines reportedly induce the chronic release of excitable neurotransmitters, inhibit the neurotransmitter uptake by astrocytes, and restrict the recycling of gamma-aminobutyric acid receptors [10,11,12]. Therefore, targeting inflammation mediated by microglia during epileptogenesis could be a strategy for preventing post-brain-insult epilepsy.

Levetiracetam (LEV) is an established second-generation anti-epileptic drug that is widely used to treat focal onset and generalized seizures [13]. We previously reported that treatment with LEV in pilocarpine (PILO)-induced SE mice largely attenuated spontaneous recurrent seizures [14]. LEV also suppresses neuroinflammation and the abnormal activation of microglia, which can attenuate subsequent post-brain-insult epilepsy [15,16]. Due to LEV′s suppression of inflammatory reactions in microglia activated by lipopolysaccharide (LPS) [16], LEV is considered to act directly on activated microglia. The synaptic vesicle protein 2A (SV2A) membrane protein is reported to be a target of LEV in the suppression of seizures [17]. However, our data clearly showed that SV2A was not expressed in microglia [16], and thus molecular targets other than SV2A have been suggested to exist in microglia. The purpose of this study was to identify a target molecule of LEV to reveal its anti-inflammatory mechanism.

## 2. Results

### 2.1. Identification of Molecular Targets of LEV in BV-2 Microglia by Cap Analysis Gene Expression Sequencing (CAGE-seq)

We have previously reported the anti-inflammatory effects of LEV [16]. LEV largely suppressed the expression of TNFα and IL-1β in the mouse microglial cell line BV-2 activated by LPS (Figure 1A). In addition, LEV suppressed the increases in TNFα and IL-1β expression observed in SE-induced epileptogenesis elicited by PILO (Figure 1B). Other inflammatory molecules increased by LPS treatment such as IL-6 and iNOS were also attenuated by LEV treatment in vitro (Appendix A) and in vivo [15]. Therefore, LEV was shown to effectively attenuate the inflammatory reaction both in vitro and in vivo.

To identify a target of LEV in the suppression of inflammatory reactions, we performed a CAGE-seq analysis. Total RNA was extracted from untreated, LPS-treated, or LPS + LEV-treated BV-2 cells, followed by the CAGE-seq analysis. Motifs whose expression was increased by LPS treatment and motifs whose activity was decreased by LEV were searched in the transcription factor-binding profiles of the JASPAR database (http://jaspar.genereg.net/, accessed on 3 October 2021, shown in Appendix A), and transcription factors that could bind to selected motifs were subsequently extracted with the motif comparison tool TOMTOM (https://meme-suite.org/meme/tools/tomtom, accessed on 3 October 2021). As a result, BATF and nuclear factor erythroid 2-related factor 2, Nrf-2 (NFE212) were identified as candidates. Furthermore, in a gene expression analysis, FosL1, musculoaponeurotic fibrosarcoma (MAF) F, and Spic were identified as candidates according to the following criteria: transcription factors whose expression levels were increased 4-fold or more by LPS and decreased by 30% or more by LEV. Therefore, we proceeded with the analysis of these five genes.

### 2.2. Suppression of FosL1 Expression and AP-1 Activity by LEV

BATF and FosL1 regulate gene transcription as subunits of the transcription factor AP-1. Therefore, the activity of the transcription factors AP-1, Nrf-2, MAF, and Spic were measured using luciferase vectors containing their consensus sequences. In BV-2 cells, the activities of AP-1 and Spic were significantly increased by treatment with LPS, and only AP-1 activity was significantly decreased by pretreatment with LEV (Figure 2A). The mRNA expression of FosL1, Spic, and MAF was increased by LPS, and FosL1 expression was significantly suppressed by LEV (Figure 2B). Treatment with LPS increased FosL1 protein levels in BV-2 cells, and pretreatment with LEV clearly reduced the increase in FosL1 levels induced by LPS (Figure 2C). Therefore, among the candidates identified by CAGE-seq, FosL1 was considered to be a likely target of LEV.

### 2.3. Increased Expression of Proinflammatory Cytokines Mediated by FosL1

AP-1 has been reported to induce the expression of inflammatory cytokines and elicit an inflammatory reaction [18]. Therefore, we investigated the relationship between FosL1 and the inflammatory reaction in BV-2 cells. When FosL1 siRNA was transfected into BV-2 cells, the increase in FosL1 expression induced by LPS was significantly suppressed (Figure 3A). The expression of TNFα and IL-1β, which were increased by LPS, were also significantly suppressed by the transfection of FosL1 siRNA (Figure 3B). Therefore, FosL1 is considered to contribute to an increase in the expression of inflammatory cytokines, at least TNFα and IL-1β, in BV-2 cells.

In the post-brain-insult epilepsy model generated by PILO-induced SE, the expression of FosL1 mRNA increased 3 h after SE and was maintained at a higher level than in naïve mice at 2 and 7 days after SE (Figure 4). Treatment with LEV tended to reduce FosL1 expression 3 h after SE and significantly reduced FosL1 expression 2 and 7 days after SE (Figure 4). These data indicate that LEV can decrease FosL1 expression in neuroinflammation accompanied by epileptogenesis.

## 3. Discussion

LEV exerts broad-spectrum antiepileptic effects and is widely used to treat focal onset and generalized seizures [13]. The synaptic vesicle protein SV2A is reported to be a target of LEV [17]. SV2A is a major vesicular protein that contains 12 transmembrane regions and can reportedly regulate vesicular exocytosis [19]. As SV2A KO mice exhibited severe seizures after the first postnatal week [20], the LEV binding to SV2A is considered to be a possible mechanism by which LEV treatment attenuates seizures [16]. LEV significantly suppressed the inflammatory reaction in BV-2 microglial cells, even though BV-2 cells do not express SV2A, indicating that LEV targets, other than SV2A, attenuate inflammation in microglia. Due to post-brain-insult epilepsy being accompanied with inflammation by brain damages, LEV has dual effects for suppressing such types of seizures; binding to SV2A to suppress exocytosis and acting on microglia to attenuate inflammation.

In this study, motif and gene expression analyses based on CAGE-seq identified FosL1 as the most likely target of LEV in microglia for attenuating inflammatory reactions.

FosL1 belongs to the Fos family, which consists of four members, Fos, FosB, FosL1 and FosL2, with all containing leucine zippers [21]. FosL1 acts as a subunit of the AP-1 transcription factor, similar to other Fos family proteins. The AP-1 transcription factors consist of members of the Jun, activating transcription factor (ATF), and MAF families, in addition to the Fos family. AP-1 homo/heterodimers are involved in a variety of biological processes, including cell proliferation, differentiation, apoptosis, and inflammation [22,23]. In this study, we showed that FosL1 plays a fundamental role in the expression of proinflammatory cytokines to elicit inflammatory reactions in microglia. FosL1 expression is elevated by B cell stimulation [24], and JunB modulates the proliferation of B cells [25]. Thus, FosL1 may be involved in regulating the inflammatory immune response along with JunB. FosL1 acts as a key downstream effector of the phosphatidylinositol 3-kinase (PI3K)/AKT signaling pathway and is responsible for the regulation of Mmp9 gene expression [26]. A further issue is the identification of FosL1 signaling involved in inflammation.

FosL1 was identified as a target of LEV by CAGE-seq in this study. LEV suppressed the expression of FosL1 and, as a result, attenuated the expression of inflammatory cytokines by reducing the activity of AP-1. Thus, LEV acts upstream of FosL1 signaling and negatively regulates FosL1 expression. Although little is known about the regulation of FosL1 expression, Lee et al. reported that A2 adenosine receptors and activated ERK were involved in events upstream of elevated FosL1 expression in the hippocampus of mice treated with kainic acid [27]. However, the expression of FosL2 also increased, indicating a lack of specificity for FosL1 in this pathway. In the present study, LEV acted only on FosL1 and did not affect the expression of FosL2 (data not shown). Further research is needed to reveal the action of LEV upstream of FosL1 expression.

Increasing evidence shows that LEV has a possibility to treat neurological disorders other than epilepsy, such as cognition of Alzheimer’s disease [28] and levodopa-induced dyskinesias in Parkinson’s disease [29]. Microglia are considered to be closely involved in the onset and progression of Alzheimer’s disease [30]. Microglia are reported to be inappropriately activated and engulf synapses in the adult brain in a microglial complement receptor-dependent mechanism, which is involved in synapse loss in Alzheimer’s disease [31]. The next step is to investigate whether the suppressive effect of LEV on microglial inflammation via AP-1 inhibition can improve neurological disorders other than epilepsy, which are accompanied by microglial activation and neuroinflammation as pathology.

## 4. Materials and Methods

### 4.1. Culture of BV-2 Cells

The mouse microglial cell line BV-2 was purchased from the Interlab Cell Line Collection (ICLC ATL03001, Genova, Italy) and cultured according to our previous report [16]. To induce a ramified microglia-like morphology, BV-2 cells were maintained with RPMI media supplemented with 0.1% FBS for 2 days. Subsequently, the cells were pretreated with 10 mM LEV for 20 min and then treated with 10 ng/mL LPS for 24 h.

### 4.2. Total RNA Extraction and Real-Time PCR

Expression levels of mRNA were determined by real-time PCR, as previously reported [32]. The primer sequences used in this study are listed in Table 1. Expression levels of mRNA were normalized to β-actin levels, and relative mRNA expression was calculated by dividing the expression levels of treated mice or cells by the expression levels of naïve mice or untreated cells.

### 4.3. Mice

The protocols for all animal experiments were approved by the Tokushima Bunri University Animal Care Committees and were performed in accordance with the National Institutes of Health (USA) Animal Care and Use Protocol. All efforts were made to minimize the number of animals used and their suffering. Six-week-old male ICR mice were purchased from Japan SLC (Shizuoka, Japan). All mice were maintained with laboratory chow and water ad libitum under a 12-h light/dark cycle.

### 4.4. PILO-Induced SE Model

The PILO-induced SE model was established according to our previous report [14]. Briefly, ICR mice (8–10 weeks old) were administered with methyl scopolamine, and then PILO (Sigma-Aldrich, St. Louis, MO, USA) was administered (290 mg/kg, i.p.). After the occurrence of five generalized convulsive seizures, diazepam was injected to terminate SE. LEV was orally administered at a dose of 360 mg/kg twice per day (at 8:30 and 17:30).

### 4.5. CAGE-seq Analysis

RNA was prepared from BV-2 cells treated with LPS or LPS+LEV using ReliaPrep RNA Miniprep Systems (Promega, Madison, WI, USA). RNA quality was assessed by using a Bioanalyzer (Agilent Technologies, Santa Clara, CA, USA) to ensure that the RNA integrity number (RIN) was over 8.0 and that the A260/A280 and 260/230 ratios were over 2.0. CAGE library preparation, sequencing, mapping, and gene expression analysis were performed by DNAFORM (Yokohama, Kanagawa, Japan). First-strand cDNAs were transcribed until reaching the 5′ end of capped RNAs and then attached to CAGE barcode tags, and these tags were sequenced using the NextSeq 500 system (Illumina, San Diego, CA, USA) and mapped to the mouse mm9 genomes using the BWA software program after discarding ribosomal RNAs. Over 20 million reads were mapped to the murine genome sequence for each sample. The data were analyzed, and the expression ratio was calculated as the log (base 2) ratio via the RECLU pipeline. The raw data were registered in the NCBI GEO database under Accession No. GSE180828.

### 4.6. Plasmid Construction and Luciferase Assay

Six repeats of the cAMP response element TGACGTCA, four repeats of the MAF recognition element TGCTGACTCAC [33], and the Spic binding site AAAAGGAAGAA [34] were cloned into the pNL3.2 vector to construct AP-1, MAF and Spic luciferase vectors. In addition to these three vectors, the pGL4.37 luciferase vector (Promega), which contains four copies of an antioxidant response element, was transfected into BV-2 cells using the Lipofectamine 2000 reagent (Invitrogen, Waltham, MA, USA). The cells were used in further experiments 24 h after transfection. Luciferase activity was measured using the Luciferase Assay System (Promega) for the pGL4 vectors, or the Nano-Glo Luciferase Assay System (Promega) for the pNL vectors, with a GloMax 20/20 Luminometer (Promega).

### 4.7. RNA Interference

Mouse FosL1 siRNA was purchased from Sigma (5′-CGACUAGAACAAACACAUUdTdT-3′). Scrambled RNA was used as a control. The mouse FosL1 siRNA and control siRNA were transfected into BV-2 cells using the Lipofectamine 2000 reagent (Invitrogen), and the cells were used in further experiments 24 h after transfection.

### 4.8. Western Blotting

BV-2 lysates were prepared with radioimmunoprecipitation assay (RIPA) buffer. The samples were loaded and separated via sodium dodecyl sulfate polyacrylamide gel electrophoresis (SDS-PAGE) using 4–20% (*w/v*) polyacrylamide gels and then transferred to polyvinylidene difluoride membranes. The blocked membranes were incubated with the following primary antibodies: anti-FosL1 (sc-11936, Santa Cruz Biotechnology, Dallas, TX, USA) and anti-β-actin (sc-47778, Santa Cruz Biotechnology). Finally, the membranes were incubated in solutions of peroxide-conjugated secondary antibodies (Thermo Fisher Scientific, Waltham, MA, USA) and then visualized using peroxide substrates (SuperSignal West Dura, Thermo Fisher Scientific).

### 4.9. Statistical Analysis

All data are presented as the mean ± standard error (S.E.). Student′s *t*-test was used to determine the significance of differences between two groups after analysis of variance (ANOVA). For the comparison of differences among three groups or more, Dunnett′s test was adopted after ANOVA. The Bonferroni method was used to correct for multiple comparisons. *p*-values < 0.05 were considered to indicate statistical significance.

## 5. Conclusions

We identified *FosL1* as a target of LEV. LEV can suppress the expression of *FosL1* under inflammatory conditions and subsequently reduce the function of AP-1 to attenuate the expression of proinflammatory cytokines in microglia. LEV reportedly binds to SV2A in neurons. The diverse activities of LEV in several types of cells might contribute to its potent antiepileptic effects.

## Figures and Tables

**Figure 1 ijms-22-10962-f001:**
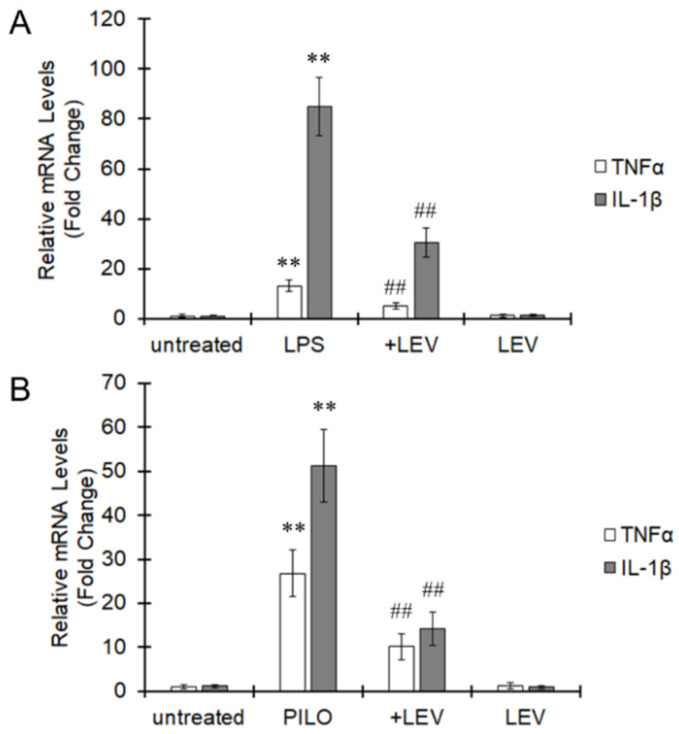
Suppression of proinflammatory cytokine expression by LEV. (**A**) BV-2 cells were pretreated with LEV for 20 min, and LPS was subsequently added for 24 h. Total RNA was extracted, and the mRNA expression of TNFα and IL-1β was evaluated by real-time PCR. The values are presented as the mean ± S.E. of 5 separate experiments. The data were analyzed using ANOVA followed by Student′s *t*-test. ** *p* < 0.01 vs. the untreated group and ^##^
*p* < 0.01 vs. the LPS-treated group. (**B**) PILO was injected into mice and SE was terminated by diazepam. LEV was repeatedly administered for 2 days. Total RNA was extracted from the hippocampus, and the mRNA expression of TNFα and IL-1β were evaluated by real-time PCR. The values are presented as the mean ± S.E. (*n* = 5 animals in each group). The data were analyzed using ANOVA followed by Student′s *t*-test. ** *p* < 0.01 vs. naïve mice and ^##^
*p* < 0.01 vs. PILO mice.

**Figure 2 ijms-22-10962-f002:**
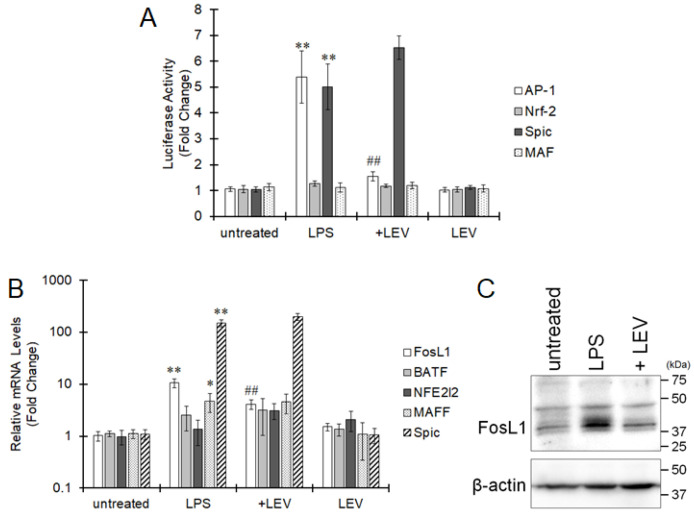
Attenuation of AP-1 activity and FosL1 expression by LEV. (**A**) Luciferase vectors were transfected into BV-2 cells with a ramified-like morphology. Cells were pretreated with LEV for 20 min and were then treated with LPS for 24 h. The cells were collected, and luciferase activity was measured. The values are presented as the mean ± S.E. of 5 separate experiments. The data were analyzed using ANOVA followed by Student′s *t*-test. ** *p* < 0.01 vs. the untreated group and ^##^
*p* < 0.01 vs. the LPS-treated group. B, C. BV-2 cells were pretreated with LEV for 20 min, and subsequently LPS was added for 24 h. (**B**) Total RNA was extracted, and mRNA expression was evaluated by real-time PCR. The values are presented as the mean ± S.E. of 5 separate experiments. The data were analyzed using ANOVA followed by Student′s *t*-test. * *p* < 0.05 and ** *p* < 0.01 vs. the untreated group and ^##^ *p* < 0.01 vs. the LPS-treated group. (**C**) Cells were collected, followed by Western blotting to assess FosL1 expression. A representative image from 3 experiments is shown.

**Figure 3 ijms-22-10962-f003:**
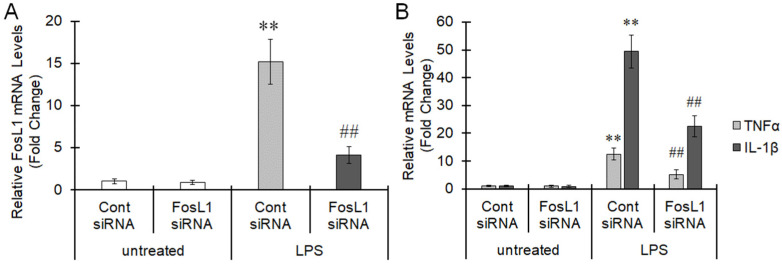
Involvement of FosL1 in microglial inflammatory reaction. Control or FosL1 siRNA was transfected into BV-2 cells with a ramified-like morphology. The cells were pretreated with LEV for 20 min, and LPS was subsequently added for 24 h. The expression of FosL1 (**A**), TNFα and IL-1β (**B**) were measured by real-time PCR. The values are presented as the mean ± S.E. of 5 separate experiments. The data were analyzed using ANOVA followed by Student′s *t*-test. ** *p* < 0.01 vs. the untreated group and ^##^ *p* < 0.01 vs. the LPS-treated group.

**Figure 4 ijms-22-10962-f004:**
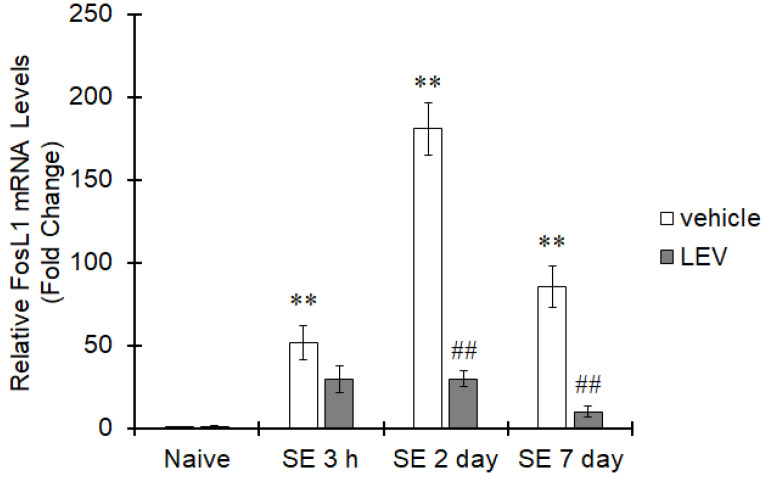
Suppression of FosL1 by LEV in a PILO was injected into mice and SE was terminated by diazepam. LEV was repeatedly administered for 7 days. Total RNA was extracted from the hippocampus at 3 h, 2 days, or 7 days after SE, and the mRNA expression of FosL1 was evaluated by real-time PCR. The values are presented as the mean ± S.E. (*n* = 5 animals in each group). The data were analyzed using ANOVA followed by Dunnett′s test or Student′s *t*-test. ** *p* < 0.01 vs. naïve mice and ^##^ *p* < 0.01 vs. PILO mice.

**Table 1 ijms-22-10962-t001:** Primers used for this study.

Target	Forward Pprimer	Reverse Primer
Mouse TNFα	ATGGCCTCCCTCTCATCAGT	CTTGGTGGTTTGCTACGACG
Mouse IL-1β	AGCTTCCTTGTGCAAGTGTCT	GCAGCCCTTCATCTTTTGGG
Mouse β-actin	CTAGGCACCAGGGTGTGATG	GGGGTACTTCAGGGTCAGGA

## Data Availability

The CAGE-seq data have been deposited in the Gene Expression Omnibus (GEO) database (accession no. GSE180828). All other data are included in the article.

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
