# Peer review of "FosL1 Is a Novel Target of Levetiracetam for Suppressing the Microglial Inflammatory Reaction"

_ijms, 2021, doi:10.3390/ijms222010962_

Round 1

Reviewer 1 Report

Niidome et al. investigated the mechanism underlying the immune modulatory effects of levetiracetam using cultures of microglial BV-2 cells. The results of the study are clearly and logically presented. I wonder if a more comprehensive evaluation of the inflammatory changes could have been performed with some additional effort. For example, is there a rationale for restricting the analyses to TNF-alpha and IL-1beta as markers of inflammation?

Do the authors expect to be any differences based on the cause of the epilepsy, for example would LEV have higher benefit in immune-mediated epilesies compared to other symptomatic (e.g., post-traumatic) epilepsies? This could be perhaps addressed in animal models and would have a great interest for clinical translation purposes.

How do the authors plan to validate their findings? Some comments on that could enrich the discussion.

Author Response

Responses to Reviewer 1

  1. Niidome et al. investigated the mechanism underlying the immune modulatory effects of levetiracetam using cultures of microglial BV-2 cells. The results of the study are clearly and logically presented. I wonder if a more comprehensive evaluation of the inflammatory changes could have been performed with some additional effort. For example, is there a rationale for restricting the analyses to TNF-alpha and IL-1beta as markers of inflammation?

Response

    Thank you for your comment. As described previously, LEV treatment significantly suppressed IL-6 and iNOS in addition to TNFα and IL-1β in the pilocarpine-induced status epilepticus model (Brain Res 1652:1, 2016). In BV-2 cells, we confirmed that LEV attenuated LPS-induced IL-6 and iNOS expression. Because AP-1 is involved in the expression of several proinflammatory genes (Cells 8:194, 2019; Biol Pharm Bull 38:1081, 2015), LEV is considered to have broad spectrum for pro-inflammatory molecules.

    We added Fig. S1 and its explanation in the Results section (P. 2, L. 26-28). Primer sequences were listed in Table S1.

  1. Do the authors expect to be any differences based on the cause of the epilepsy, for example would LEV have higher benefit in immune-mediated epilepsies compared to other symptomatic (e.g., post-traumatic) epilepsies? This could be perhaps addressed in animal models and would have a great interest for clinical translation purposes.

Response

    Thank you so much for the comment. LEV is an established second-generation antiepileptic drug that is widely used in patients with either generalized or partial epilepsy. LEV is considered to act through an interaction with the synaptic vesicle protein 2A (SV2A) to suppress excess excitation of neurons. On the other hand, we showed that LEV suppressed microglial inflammatory reaction in a SV2A independent manner because microglia do not express SV2A proteins (Neurosci Lett 708:134363, 2019). Because post-brain-insult epilepsy is accompanied with inflammation by brain damages, LEV has dual effects for suppressing such types of seizures; binding to SV2A to suppress exocytosis and acting on microglia to attenuate inflammation.

    We added this discussion in the Discussion section (P. 5, L.16-19).

  1. How do the authors plan to validate their findings? Some comments on that could enrich the discussion.

Response

    Thank you for the comment. We think that one of the future plans is the expansion of LEV usage to other neurological disorders. We discussed on Alzheimer’s disease and microglial activation in the Discussion section (P. 5, L.45-53).

Reviewer 2 Report

The authors investigated the target of levetiracetam (LEV) on microglia during neuroinflammation. They have reported the anti-inflammatory effects and anti-seizure of LEV  before. This study is the extension of their previous work to identify the underlying molecular mechanisms. Overall, the manuscript is straightforward and the data is clearly presented. I only have several minor points:

  1. In the last paragraph of the introduction, the authors state "molecular targets other than SV2A have been suggested to exist in microglia". Please add references here.
  2. In the result section 2.1, I would suggest listing the genes analyzed in CAGE-seq, TOMTOM in like Excel file as supplementary data. The readers may be interested in other gene analyzed in this study.
  3. Does the neuroinflammation come back after stopping the treatment of LEV? Does LEV okay for longterm treatment? Except for the SE model, do you have any references to apply LEV in other neurodegenerative diseases, like AD, PD? You may consider adding some more information in the discussion section.

Author Response

Responses to Reviewer 2

Authors investigated the target of levetiracetam (LEV) on microglia during neuroinflammation. They have reported the anti-inflammatory effects and anti-seizure of LEV before. This study is the extension of their previous work to identify the underlying molecular mechanisms. Overall, the manuscript is straightforward and the data is clearly presented. I only have several minor points:

  1. In the last paragraph of the introduction, the authors state "molecular targets other than SV2A have been suggested to exist in microglia". Please add references here.

Response

    Thank you very much for the indication. We have shown that microglial cell line BV-2 did not express SV2A (Neurosci Lett 708:134363, 2019). Based on this result, we suggested that there is other target of LEV than SV2A in microglia. We improved the sentence (P.2. L. 16-18).

  1. In the result section 2.1, I would suggest listing the genes analyzed in CAGE-seq, TOMTOM in like Excel file as supplementary data. The readers may be interested in other gene analyzed in this study.

Response

    Thank you for the comment. We already uploaded our CAGE-seq raw data in the NCBI GEO database under Accession No. GSE180828. Thus, readers can get the information required from the database. The motifs that we found in JASPAR database were listed as Table S2.

  1. Does the neuroinflammation come back after stopping the treatment of LEV? Does LEV okay for longterm treatment? Except for the SE model, do you have any references to apply LEV in other neurodegenerative diseases, like AD, PD? You may consider adding some more information in the discussion section.

Response

    Thank you for the comment. We do not confirm the LEV withdrawal but based on our experiments, LEV treatment was effective on suppression of seizures at least during the 28 days after SE (Brain Res 1608:225, 2015). Therefore, it is considered that LEV can suppress the onset of seizures for long-term without desensitization.

   Also, thank you for the comment on neurodegenerative diseases. We added the discussion about LEV and Alzheimer’s disease in the Discussion section (P.5, L.45-53).